# Insights about Modelling Environmental Spatiotemporal Actions in Thermal Analysis of Concrete Dams: A Case Study

**Noemi Schclar Leitão** *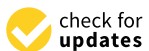 and **Sérgio Oliveira** 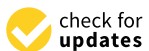

Laboratório Nacional de Engenharia Civil (LNEC), Av. do Brasil, 101, 1700-066 Lisbon, Portugal
* Correspondence: nschclar@lnec.pt

**Abstract:** In order to conduct thermal analysis of concrete dams, it is necessary to assess and validate the spatiotemporal representations used for modeling the solar radiation and the water temperature boundary conditions. To illustrate this procedure, the thermal analysis of a concrete multiple-arch dam is presented. The article starts by providing an overview of the problem before focusing explicitly on the estimation of solar radiation distribution. Within this section, a comparison between the solar irradiance computed on the downstream face of the dam with or without considering the beam radiation shading at different times of the year is presented. This is followed by an analysis of the seasonal behavior of the water temperature of the dam's reservoir based on measured data. After calibrating an empirical/statistical law based on temperatures measured at different depths, it is compared with the values estimated by a hydrodynamic model and some temperature profiles measured upstream of the dam. Finally, the article compares the results obtained with the thermal analysis versus the temperature measured by thermometers installed in the concrete.

**Keywords:** heat transfer analysis; finite element analysis; environmental actions; solar radiation; water temperature; multiple arch dam

## 1. Introduction

Numerical modeling is an important tool for supporting the continuous monitoring of large concrete dams by helping the timely detection of abnormal behavior, the prediction of the service life, and the implementation of an effective maintenance strategy. In this context, the simulation of the environmental actions requires special attention due to their substantial impact on dams' reactions (displacements, strains, and stresses). In this sense, it is important to note that temperature has a dual role: one as an imposed deformation, the other as a critical factor for hydration reaction, creep, and expansive reactions.

However, while standards provide comprehensive information about mechanical loadings, environmental actions remain poorly defined, resulting in a variety of different criteria for selecting the main phenomena involved in thermal analysis [1].

As in any modeling process, many different aspects will define the features and events to be considered in the thermal analysis. The objective of the coupled thermo-mechanical analysis, the age of concrete, and the required level of agreement between reality and simulation outcomes will determine the degree of complexity of the thermal model.

From the review of the computational aspects of the thermo-mechanical modeling of arch dams presented by Salazar et al. [1], it can be seen that some phenomena have been considered in all relevant works published over the last decade. The consideration of the heat exchange by convection between the surface of the structure and the air and the water temperature as imposed boundary conditions at the upstream face of the dam is a common feature of all analyses. Other dam-air exchange fluxes, such as solar (short wave) and atmospheric (long wave) radiation, reflected solar and atmospheric radiation, evaporation, or night cooling, have received less attention or have simply been neglected.

Moreover, some phenomena present a spatiotemporal distribution that has to be determined before using it as a boundary condition, such as the case of solar radiation and water temperature.

In the case of solar radiation, the spatiotemporal distribution is governed by geometrical concepts, making it easy to compute the seasonal and hourly changes of solar radiation over the dam [2,3]. Shadow effect, instead, requires the use of labor-intensive and time-consuming shadow detection procedures. From the pioneering work of Jin et al. [4] to the most recent works [5,6], authors have used the ray-tracing algorithm to define the shaded area caused by the surrounding terrain and the dam itself. An exception is the work of Santillán and coworkers, who computed the shading of the dam by recourse to the projection of the different surfaces in the direction of the sun rays [7,8]; however, later on, they also adopted the ray-tracing algorithm [9,10].

In the case of the water temperature, the spatiotemporal distribution is governed by physics, mainly heat transfer and heat transport. The total heat budget for a water body includes the effect of atmospheric heat exchange at the air-water interface, surface and subsurface inflows and outflows (including precipitation and groundwater), and heat transfer through the bed. However, due to a scarcity of or expense in collecting field data, together with the complexity of the computations, simplified models have usually been adopted. Such models range from solving one-dimensional thermal diffusivity equations to empirical solutions [11].

The purpose of this work is to evaluate and validate the representations adopted for these two spatiotemporal fields in the thermal analysis of a multiple-arch dam. Despite the numerous articles written about thermal analysis of concrete dams, the comparison of different approaches to represent the spatiotemporal fields is scarce and mainly refers to solar radiation being considered or not. Therefore, in the research reported here, several comparisons will be presented. The article starts by providing an overview of the problem before focusing explicitly on the estimation of solar radiation distribution. Within this section, a comparison between the solar irradiance computed on the downstream face of the dam with or without considering the beam radiation shading at different times of the year is presented. This is followed by an analysis of the seasonal behavior of the water temperature of the dam's reservoir based on measured data. After that, an empirical/statistical law is calibrated with the temperatures measured at different depths. The adopted law is compared with the values estimated by a hydrodynamic model and some temperature profiles measured upstream of the dam. Finally, the article compares the results obtained with the thermal analysis versus the temperature measured by thermometers installed in the concrete.

## 2. Dam Description

Aguieira Dam is a concrete multiple-arch dam on the Mondego River located in Coimbra District, Portugal. Three double curvature arches and two central buttresses form the multiple arch dam. Each buttress has a spillway (Figure 1).

The dam's construction began in 1972 and was completed in 1981. Apart from power production, the dam is also used for flood control, water supply, and irrigation.

The maximum height above the foundation is 89 m, the crest length is 400 m, and the crest elevation is 126.65 m.

The foundation is a schistose rock mass with greywacke alternations.

The hydroelectric power plant is located at the base of the central arch (Figure 2). It is equipped with three reversible Francis pump turbines.

Instruments of various types were installed in the dam as part of its monitoring system; they supply continuous information on displacements, deformations, seepage, uplift pressures, and ambient and internal temperatures.

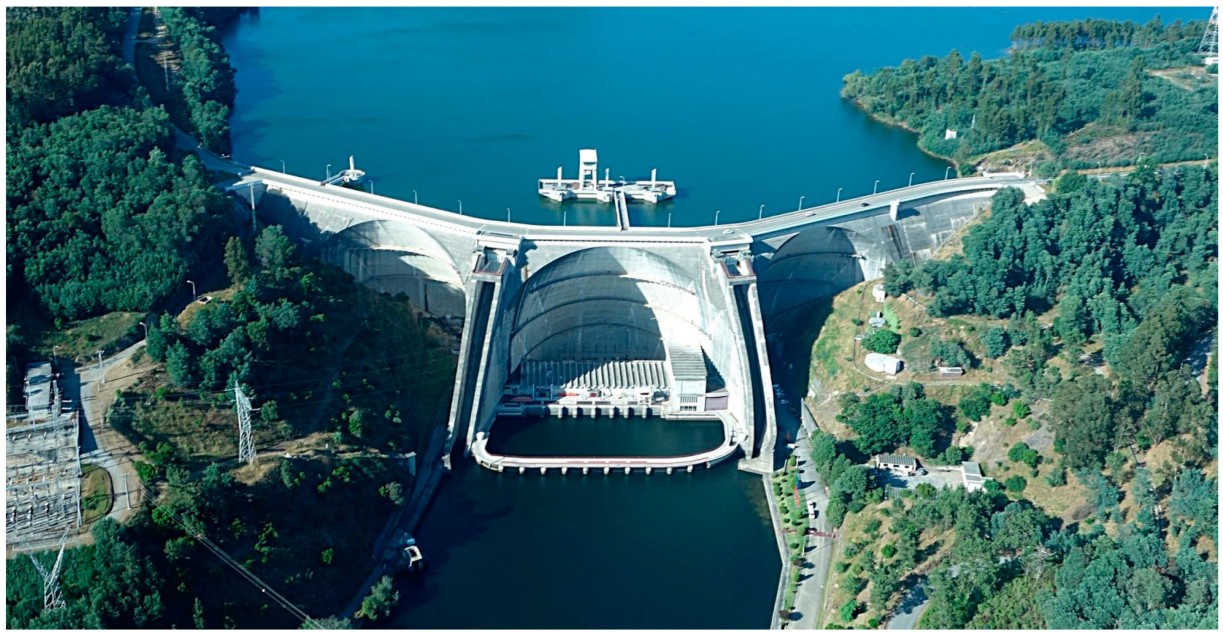

**Figure 1.** Downstream view of Aguieira Dam [12].

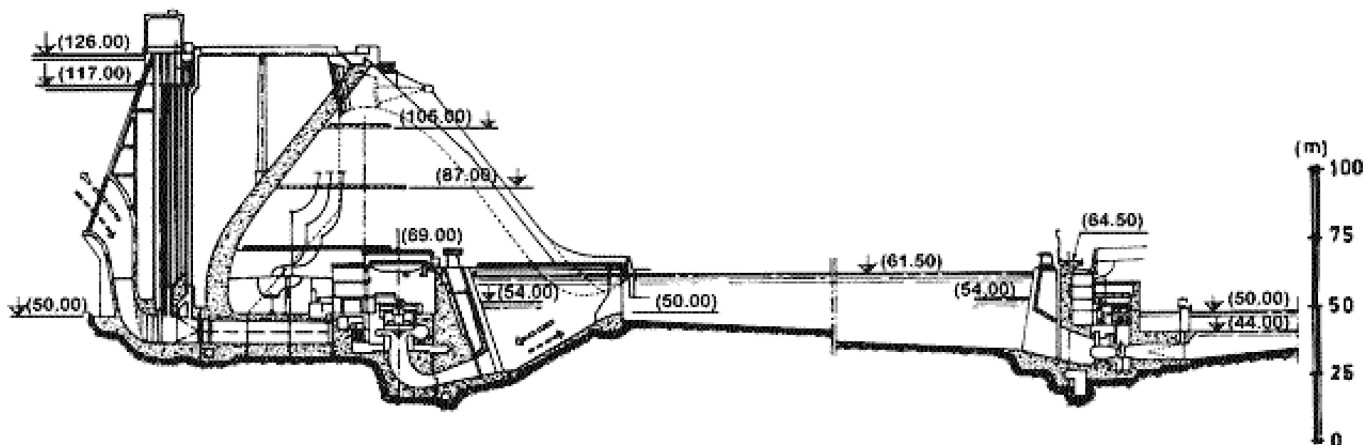

**Figure 2.** Cross-sectional view through the axes of one generating unit.

### 3. Governing Equations

The thermal analysis is governed by the transient heat conduction equation

$$\frac{\partial}{\partial x}\left[k_x\frac{\partial T}{\partial x}\right] + \frac{\partial}{\partial y}\left[k_y\frac{\partial T}{\partial y}\right] + \frac{\partial}{\partial z}\left[k_z\frac{\partial T}{\partial z}\right] + G = \rho\,c\frac{\partial T}{\partial t} \tag{1}$$

with the following boundary conditions

$$T = \overline{T} \qquad \text{in } \Gamma_T \tag{2}$$

$$k_x\frac{\partial T}{\partial x}\,l + k_y\frac{\partial T}{\partial y}\,m + k_z\frac{\partial T}{\partial z}\,n + q_c + q_r + q_q = 0 \quad \text{in } \Gamma_q \tag{3}$$

and the following initial condition

$$T = T_o \qquad \text{in } \Omega \text{ for } t = t_o \tag{4}$$

where $t$ is the time; $T$ is the temperature; $k_x$, $k_y$ and $k_z$ are the thermal conductivities; $G$ is the internally generated heat per unit of volume and time; $\rho$ is the material density; $c$ is the specific heat; $\overline{T}$ is the temperature at the boundary $\Gamma_T$; $q_c$ is the heat flux due to convection; $q_r$ is the heat flux due to atmospheric (long wave) radiation and $q_q$ is the solar (short wave) radiation at the boundary $\Gamma_q$; $l$, $m$, and $n$ are the direction cosines; and $T_o$ is the temperature at the time $t_o$. It is noteworthy that the convention in expression (3) is positive when heat flux flows outwards from the body [13]. When the material is isotropic, the thermal conductivity is the same in all directions, that is $k_x = k_y = k_z = k$.

The heat exchange by convection between the surface of the structure and the air depends on wind speed and air temperature. The heat gain or loss from a surface due to convection is given by Newton's law

$$q_c = h_c \left( T - T_a \right) \tag{5}$$

where $T_a$ is the air temperature and $h_c$ is the convection heat transfer coefficient, which is a function of the wind speed.

The temperature difference between the surface of the structure and the air gives origin to electromagnetic radiation, which is measured by the Stefan–Boltzmann law

$$q_r = \varepsilon\, \sigma\, \left( T^4 - T_a^4 \right) \tag{6}$$

where $\varepsilon$ is the emissivity of the surface, and $\sigma$ is the Stefan–Boltzmann constant given as $5.669 \times 10^{-8}$ W (m$^2$ K$^4$). When $T$ and $T_a$ are close, which is the case in civil engineering structures, it is possible to rewrite (6) in a quasi-linear form

$$q_r = h_r(T - T_a) \tag{7}$$

where $h_r$ is the radiation linear coefficient defined as follows

$$h_r = \varepsilon\, \sigma\, \left( T^2 - T_a^2 \right) (T - T_a) \tag{8}$$

Combining the contribution of both heat transfer mechanisms, i.e., convection and radiation, it is possible to define a new coefficient called the total thermal transmission coefficient, $h_t$. This new coefficient, in essence, is a convection heat transfer coefficient that is updated to consider radiation.

The solar radiation boundary condition is given by

$$q_q = a\, I_T \tag{9}$$

where $a$ is the absorption coefficient and $I_T$ is the solar irradiance.

## 4. Finite Element Model

### 4.1. Finite Element Formulation

The transient Equation (1) subjected to the appropriate boundary and initial conditions given by Equations (2)–(4) is solved by applying the standard Galerkin Finite Element Method (FEM).

Following the formulation given in [13], the temperature is approximated over space using FEM approximation

$$\mathrm{T} = \sum_{i=1}^{n} N_i\, T_i \tag{10}$$

where $N_i$ are the shape functions, $n$ is the number of nodes in a domain, and $T_i$ is the time-dependent nodal temperature. Meanwhile, the time is integrated using the Finite Difference Method (FDM).

After applying the FEM in space and introducing the θ-method for time integration using FDM, the resulting fully discretized system of linear algebraic equations can be written as

$$([C] + \theta \, \Delta t \, [K])\{T\}^{n+1} = ([C] - (1-\theta) \, \Delta t \, [K])\{T\}^n + \Delta t\left(\theta\{f\}^{n+1} + (1-\theta)\{f\}^n\right) \quad (11)$$

where $[C]$ is the capacitance matrix

$$[C] = \int_\Omega \rho \, c \, [N]^T [N] d\Omega \tag{12}$$

$[K]$ is the heat stiffness (conduction and convection) matrix

$$[K] = \int_\Omega [B]^T [D] [B] \, d\Omega + \int_{\Gamma_q} h_t [N]^T [N] d\Gamma_q \tag{13}$$

and $\{f\}$ is the total load heat vector

$$\{f\} = \int_\Omega G \, [N]^T \, d\Omega - \int_{\Gamma_q} q_q [N]^T d\Gamma_q + \int_{\Gamma_q} h_t T_a [N]^T d\Gamma_q \tag{14}$$

where the first integral takes into account the internal heat generation, the second integral takes into account the solar radiation heat flow, and the third integral takes into account the convection and atmospheric radiation heat transfer.

In the above integrals, $[N]$ refers to the shape function matrix, $[B]$ refers to the gradient matrix of the shape functions and $[D]$ is the thermal conductivity matrix.

### 4.2. Finite Element Mesh

The finite element model representing the multiple arch dam and an appropriate volume of the foundation used in the analysis is shown in Figure 3. The model has 3784 quadratic 20-node brick finite elements, 1448 representing the dam body, and 2336 representing the foundation, corresponding to a total number of 20,065 nodal points.

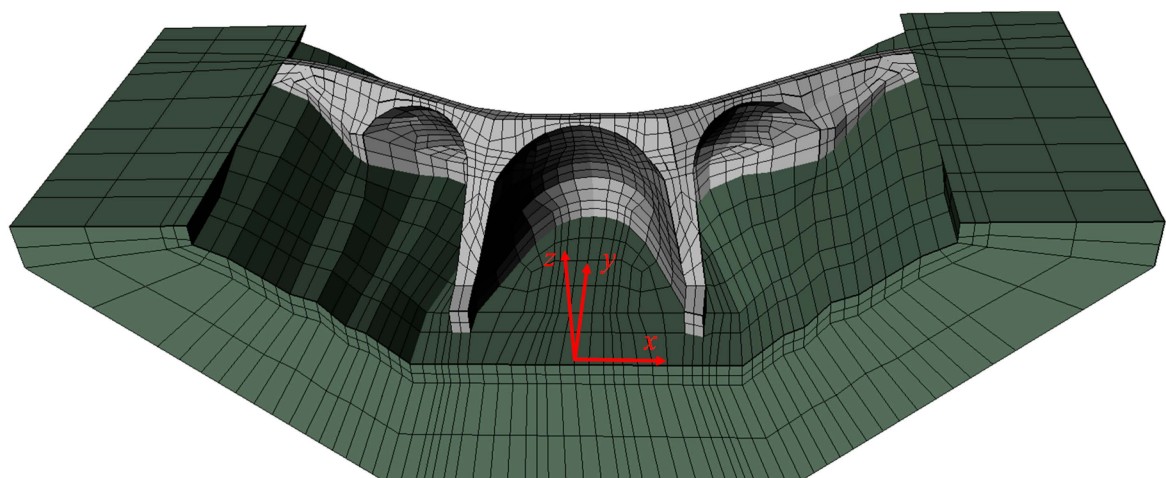

**Figure 3.** Finite element model of the dam.

The coordinate system considers the positive $x$-direction toward the left bank, the positive $y$-direction toward the upstream, and the positive $z$-direction pointing to the Zenith. The global azimuth, which is the angle from the reference direction (in this work, the south) to the $y$-direction, measured clockwise around the surface's horizon, is $\psi_y = 219°$.

## 5. Thermal Properties

The thermal properties of Aguieira Dam were estimated for an average composition of the concrete by Telles [14] using the method reported by the U.S. Bureau of Reclamation in [15]. This method is based on the mix proportions and petrographic composition of aggregates. It assumes that each material the concrete is composed of contributes to the conductivity and specific heat in proportion to the amount of the material present in the concrete, given $k$ = 2.6 W/(m °C) and $c$ = 955 J/(kg °C). Due to the lack of information, similar values were adopted for the rock foundation.

For both materials, the value of 2400 kg/m$^3$ was adopted for the density.

Regarding the absorption coefficient, it was adopted for the concrete with a value of 0.6 and no solar radiation effect for the rock mass foundation.

## 6. Convection Heat Transfer

The daily air temperature variation was estimated based on the minimum and maximum daily temperatures measured at the dam site. For each day, it was assumed that the extreme temperatures occurred 12 h apart, at 03:00 h and 15:00 h for the minimum and maximum temperatures, respectively, and a linear variation was assumed between them.

The convective heat transfer coefficient $h_c$ was estimated using the expression given by Brown and Marco (1958) (as cited in [16])

$$h_c = 0.055 \frac{k_f}{L} \left( \frac{L \, V_w \rho_f}{\mu_f} \right)^{0.75} \quad [\text{W}/(\text{m}^2\text{K})] \tag{15}$$

where $k_f$, $\rho_f$ and $\mu_f$ are the thermal conductivity, density, and absolute viscosity of air, which correspond to the values of 0.026 W/(m K), 1.2 kg/m$^3$, and $1.8 \times 10^{-5}$ kg/(m s), $V_w$ is the average wind speed in m/s, and $L$ represents the size of the considered flat surface, for which Silveira [16] adopted the value of 0.60 m. Then, considering an average wind speed $V_w$ = 2.8 m/s (corresponding to 10 km/h), the convection coefficient results in $h_c$ = 14.6 W/(m$^2$ K).

Regarding the linearized radiation coefficient, for the range of temperature values registered in Portugal, Silveira [16] adopted a constant value of 5 W/(m$^2$ K). Therefore, a constant value for the total thermal transmission coefficient $h_t$ = 20 W/(m$^2$ K) was applied to the whole model.

## 7. Solar Radiation

### 7.1. The Path of the Sun across the Celestial Sphere

The apparent motion of the sun, caused by the rotation of the earth about its axis, changes the angle at which the beam component of the sunlight will strike the earth. From the point of view of an observer, the sun appears to move along with the celestial sphere on any given day but follows different circles at different times of the year: most northerly at the June solstice and most southerly at the December solstice. At the equinoxes, the sun's path follows the celestial equator. Figure 4 shows the sun's paths relative to the dam position for the equinoxes and solstices.

### 7.2. Solar Position

The course of the sun can be described by two angles: the solar altitude above the horizontal $\alpha_s$ and the solar azimuth measured from the south $\psi_s$ (Figure 5). Both angles are a function of the sun's declination $\delta$, the earth's latitude $\phi$, and the solar hour angle $\omega$ defined as the angular displacement of the sun east or west of the local meridian due to the rotation of the earth.

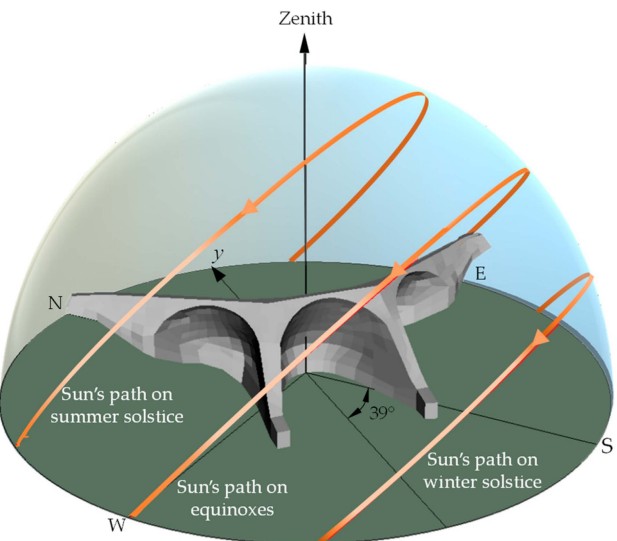

**Figure 4.** The sun's paths across the sky as seen from the dam.

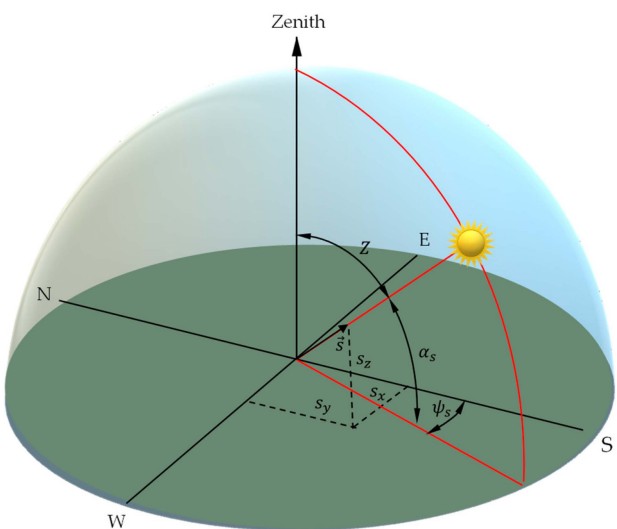

**Figure 5.** Sun's position, spherical and Cartesian coordinates.

According to [17], these angles can be calculated as

$$\sin \alpha_s = \cos Z = \sin \phi \sin \delta + \cos \phi \cos \delta \cos \omega \tag{16}$$

$$\psi_s = \text{sign}(\omega) \left[ \cos^{-1} \left( \frac{\cos \delta \cos \omega - \cos Z \cos \phi}{\sin Z \sin \phi} \right) \right] \tag{17}$$

$$\omega = 15(AST - 12) \tag{18}$$

where $AST$ is the apparent solar time. The conversion between local standard time $LST$ and solar time involves the equation of time $ET$ and a longitude correction, which corresponds to four minutes of time per degree difference between the local longitude $LON$ and the longitude of the local standard meridian $LSM$ for the time zone

$$AST = LST + \frac{ET}{60} + (LON - LSM)/15 \tag{19}$$

In some applications, it is helpful to replace the spherical coordinates with Cartesian coordinates. Considering a coordinate system where the positive $x$-direction is toward the

west, the positive $y$-direction is toward the south, and the positive $z$-direction is pointing to the Zenith, the transformation from spherical coordinates to Cartesian coordinates is given by

$$\vec{s} = \left\{ \begin{array}{c} \sin \psi_s \cos \alpha_s \\ \cos \psi_s \cos \alpha_s \\ \sin \alpha_s \end{array} \right\} \tag{20}$$

where $\vec{s}$ is a unit vector representing the solar ray direction.

### 7.3. Angle of Incidence

The intensity of solar radiation on a surface depends upon the angle at which the sun's rays strike the surface. The intensity is proportional to the cosine of the angle between the solar rays $\vec{s}$ and the surface's normal vector $\vec{n}$. This angle is called the angle of incidence $\alpha$ and is given by [18]

$$\cos \alpha = A \sin \delta + B \cos \omega \cos \delta + C \sin \omega \cos \delta \tag{21}$$

with

$$\begin{aligned} A &= \cos Y \sin \phi - \sin Y \cos \phi \cos \psi \\ B &= \cos Y \cos \phi + \sin Y \sin \phi \cos \psi \\ C &= \sin Y \sin \psi \end{aligned} \tag{22}$$

where $Y$ is the tilt angle, and $\psi$ is the azimuth of the surface.

If the right-hand side of Equation (21) is negative, the sun's rays will not strike the front side of the surface, and therefore it will be shaded.

### 7.4. Solar Radiation Components

Solar radiation at the earth's surface consists of two components: beam and diffuse solar radiation.

Beam or direct radiation refers to radiation that comes in a beam directly from the sun. Diffuse solar radiation is the scattered solar radiation from throughout the sky. Although diffuse radiation is most intense near the sun, a good approximation is to assume that it is isotropic, that is, uniformly distributed in all directions.

A third radiation component is the radiation reflected from the ground and from other surrounding objects onto a sloped surface.

According to the Liu–Jordan theory [19], the hourly solar radiation $I_T$ that reaches a surface tilted at an angle $Y$ (in reference to a horizontal plane) is described by the equation

$$I_T(t) = I_b(t) R_b(t) + I_d(t) R_d + [I_b(t) + I_d(t)] \rho_o R_r \tag{23}$$

where $I_b$ and $I_d$ are, respectively, the beam and the diffuse radiation intercepting on a horizontal surface, $\rho_o$ is the surface (ground) reflectance and $R_b$, $R_d$ and $R_r$ are, respectively, the correction factors for beam, diffuse, and reflected radiation

$$R_b = \frac{\cos \alpha}{\cos Z} \tag{24}$$

$$R_d = \frac{1 + \cos Y}{2} \tag{25}$$

$$R_r = \frac{1 - \cos Y}{2} \tag{26}$$

### 7.5. Solar Irradiance

When no measurements are available, solar irradiance can be estimated using statistical values obtained in nearby solar radiation stations or using solar radiation models.

In this case, the solar irradiance can be estimated using the statistical values obtained from the data measured in the solar radiation station of Coimbra, which is situated about 23 km (beeline) downstream of the dam and reported by Silveira [16]. In order to use this data, given in graphical form, an exponential function was fitted

$$\frac{I_b}{\cos Z} = I_o \exp(-0.942 + 0.5195 \cos Z) \tag{27}$$

where $I_o$ is the solar constant (1367 W/m$^2$).

In fact, the solar irradiance measured in the solar radiation station of Coimbra corresponds to the global horizontal irradiation, which is the sum of the beam and diffuse radiations. However, as no data is available to decompose the radiation, all the measured irradiance was considered beam radiation, and the diffuse radiation was neglected.

It is also worth noting that the function represents average measured values, which are already affected by average cloud cover conditions.

Alternatively, it was used the radiative model reported by Kumar et al. [20]. Despite its low ranking in overall performance when compared to other models [21], the simplicity of this approach makes it attractive in areas where great accuracy is not necessary, which is the case of civil engineering applications, given the degree of uncertainty in the characterization of the different variables involved in this type of analysis. Dam applications of this model can be seen in [2,22].

### 7.6. Shading of Beam Radiation

As the sun moves through the sky from east to west, the orientation of shadows cast by the dam's own geometry or the slopes changes. Hence, it is important to determine which part of the surface will be shaded at any particular time.

There are different ways to determine the shading of beam radiation. One of the most popular methods is the ray-tracing technique. The basic idea of this method is to start a ray at the object and send it to the sun. If this backward ray reaches the sun without hitting any object along its way, then the beam radiation strikes the object; otherwise, it will be in shadow. The problem of the intersection of a ray with an object is one of the classical problems in the field of computer graphics, where many algorithms for ray tracing have been developed. For this work, the ray-triangle intersection algorithm presented in [23] was adopted.

In order to use this algorithm, the quadrilateral mesh formed by the associated surface elements on the dam foundation's exposed surface must be transformed into a triangular mesh. This is carried out by subdividing each quadrilateral into two linear triangles, considering the nodes only at vertices.

It is worth noting that before applying the algorithm, the ray unit vector defined in (20) must be rotated anticlockwise around the *z*-axis by the reference azimuth angle $\psi_y$

$$\vec{r} = [R]\, \vec{s} \tag{28}$$

with

$$[R] = \begin{bmatrix} \cos \psi_y & -\sin \psi_y & 0 \\ \sin \psi_y & \cos \psi_y & 0 \\ 0 & 0 & 1 \end{bmatrix} \tag{29}$$

Although this shadow detection procedure is time-consuming, it is very easy to implement. Moreover, as the shading only depends on the time of the year, the shadow detection must be carried out only once, and the results are stored in a table of zero and one values for each element in order to use during the computation.

Figure 6 compares the solar irradiance computed on the downstream face of the dam with or without considering the beam radiation shading for the summer and winter solstices and the equinox at 12 h (solar hour). To indicate the sun's position, we adopted the position of the downstream thermometer T3 as a receiver. As shown in the figure, the

complex geometry of the dam frequently intercepts the solar rays, creating shaded areas in the downstream face of the dam.

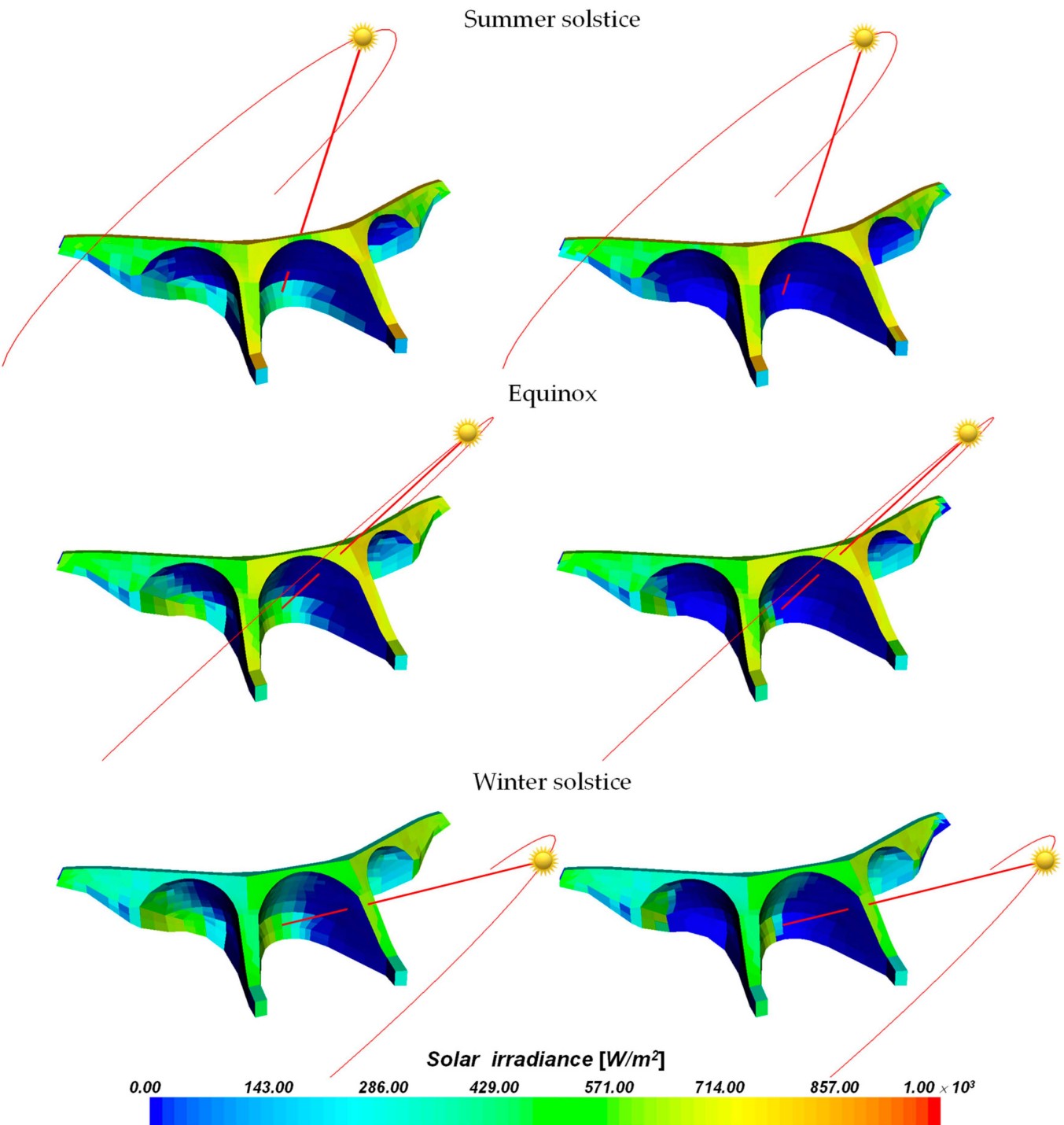

**Figure 6.** Comparison of the solar irradiance on the downstream face of the dam at 12 h (solar time) during the summer solstice, the winter solstice, and the equinox.

## 8. Reservoir Water Temperature

For the study of the water quality of Aguieira Dam, Coelho [24] measured the temperature of the reservoir at different depths during 1998. These monthly variations of water temperature are illustrated in Figure 7.

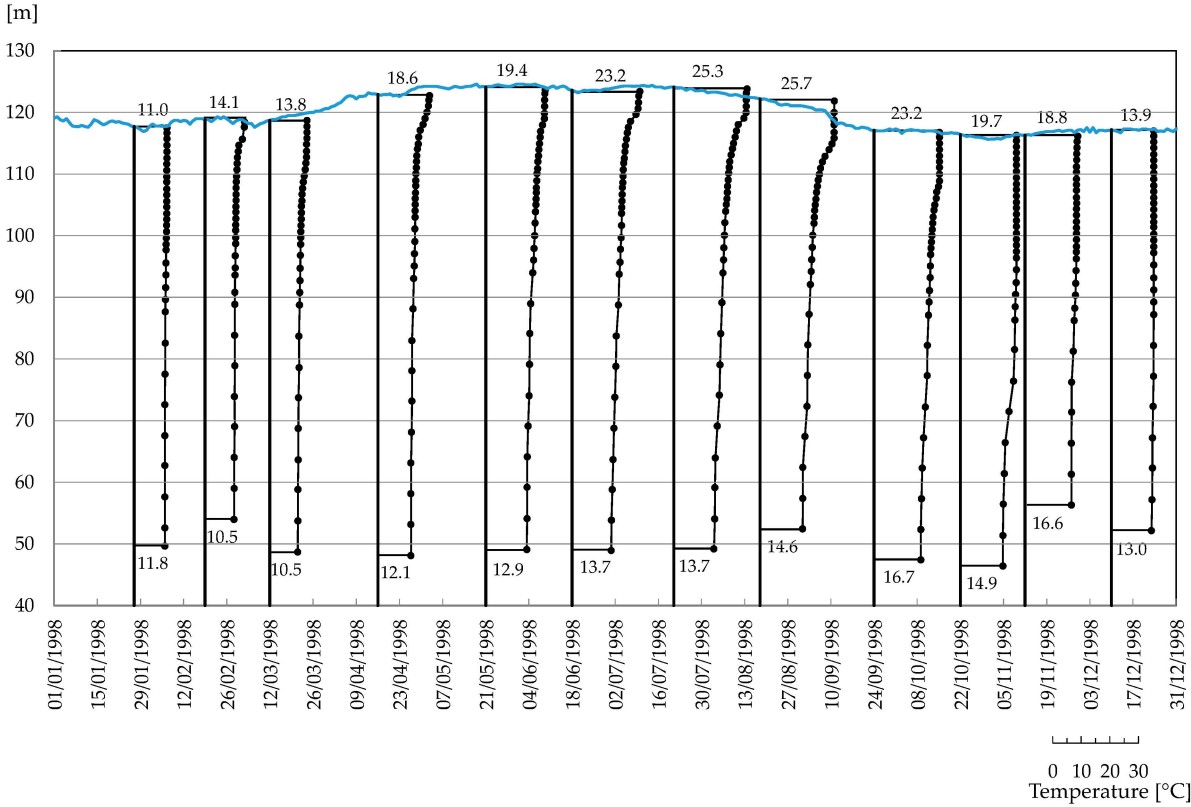

**Figure 7.** Water temperature profiles upstream of Aguieira Dam during 1998.

As can be seen from the temperature profiles, in the transition from winter to spring, the reservoirs present an almost uniform distribution of temperature of the water column due to the mixing of the entire water column. This phenomenon is called turnover. Then, the surface layer of the reservoir begins to warm in response to the increasing intensity of solar radiation. Over the summer months, continued input of solar radiation accompanied by wind mixing keeps on with the warming of the surface layer, making it increasingly less dense than the cool water below. This process results in the division of the reservoir into three layers of water, known as the epilimnion, metalimnion, and hypolimnion (Figure 8). The epilimnion is the warmer upper layer and is typically well mixed. The metalimnion is the middle layer, which is the layer where the temperature decreases rapidly with increasing depth. The hypolimnion is the bottom layer of colder water, extending to the bed of the reservoir. As the summer turns to fall, the surface water cools and sinks, mixing the epilimnion down towards the hypolimnion and reducing the metalimnion as the temperatures and densities of the epilimnion and hypolimnion become more similar. This is the fall turnover of the reservoir. As cooling accelerates, convective mixing rapidly deepens the isothermal zone, and the reservoir gradually progresses toward an isothermal condition. Because Portugal does not get particularly cold in winter, reservoirs do not freeze over winter. Therefore, reservoirs mix throughout the fall, winter, and spring and stratify in the summer.

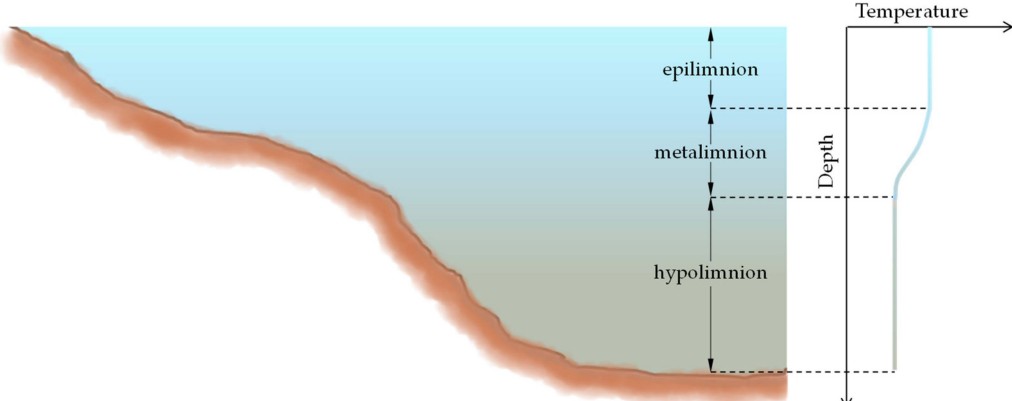

**Figure 8.** Schematic summer thermal stratification of the reservoir showing epilimnion, metalimnion, and hypolimnion and associated thermal profile.

It is worth mentioning that, besides external heat inputs, reservoir temperatures are also influenced by inflows and outflows because they affect the residence time of water within the reservoir [25]. Therefore, the simulation of temperature dynamics in reservoirs involves accounting for the various inputs and outputs of thermal energy, which cause the temperature of a parcel of water to either increase or decrease. Energy can be exchanged at the air-water interface and at the bed-water interface and can be redistributed vertically or gained or lost horizontally via advection and mixing processes. To account for these inputs and outputs of thermal energy, simulation models typically use a heat budget approach [26].

Since water temperature plays a critical role in the functioning of aquatic ecosystems, a variety of models have been developed to assess the water quality and support management of aquatic resources. High-fidelity hydrodynamic and water quality numerical simulation models, like the CE-QUAL-W2 and Environmental Fluid Dynamics Code (EFDC), can faithfully simulate the distribution of water temperature. However, the cost of such models' calculations is high, and they typically take several hours to several days to complete. To address this problem, data-driven models have been used to replace numerical models. Such data-driven models are trained with data generated with numerical models. Even though the performance of these data-driven models is encouraging, they still present some limitations nowadays [27].

By contrast, in structural engineering, the estimation of the reservoir temperature is usually simplified using unidirectional vertical thermal diffusion analytical solutions or empirical expressions [11].

In this work, we adopted the empirical/statistical approach presented by Bofang [28]

$$T(y,d) = T_m(y) - A(y) \cos\left\{ \frac{2\pi}{365}[d - d_o(y)] \right\} \tag{30}$$

with

$$T_m(y) = c + (T_s - c)\exp(-e_1 y) \tag{31}$$

$$A(y) = A_o \exp(-e_2 y) \tag{32}$$

$$d_o(y) = \tau_o + [e_3 - e_4 \exp(-e_5 y)]\frac{365}{12} \qquad \text{[days]} \tag{33}$$

where $y$ is the depth of the water, $d$ is the fractional day of the year, $T_m(y)$, $A(y)$ and $d_o(y)$ are the annual mean temperature, the amplitude of annual variation, and the phase difference of water temperature at depth $y$, $\tau_o$ is the time for maximum air temperature, and $T_s$, $A_o$, $c$ and $e_1$ to $e_5$ can be obtained through the monitored temperatures.

After obtaining the mean harmonic behavior of each of the twelve thermometers installed at the upstream face of Aguieira Dam, the following values were fixed $\tau_o = 24.35$ days, $T_s = 17.6\,°C$, $A_o = 7.3\,°C$, $c = 10.33$ days, $e_1 = 0.01$, $e_2 = 0.015$, $e_3 = 2.02$ month, $e_4 = 2.10$ month, and $e_5 = 0.085$, as shown in Figure 9.

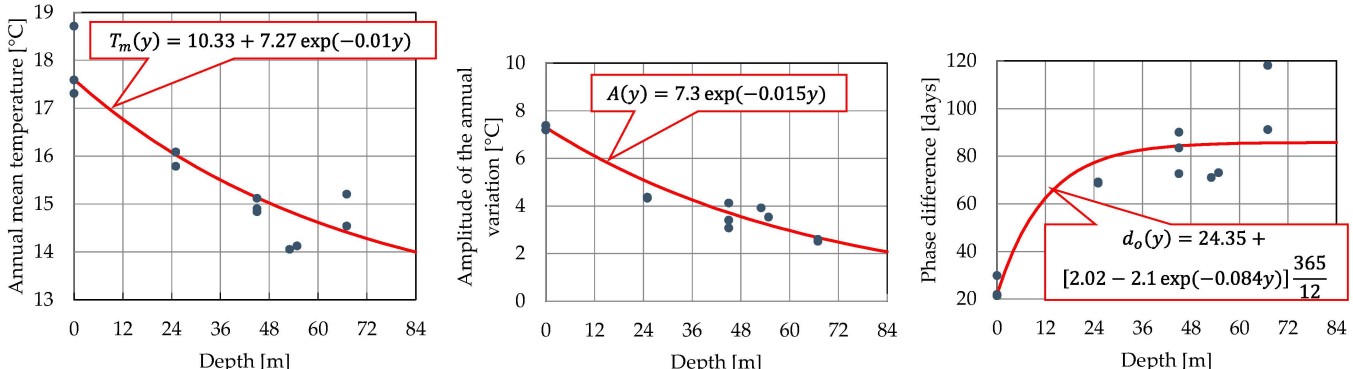

**Figure 9.** Mean observed values and empirical water temperature variations of $T_m(y)$, $A(y)$ and $d_o(y)$ for the Aguieira dam reservoir.

Graphical results comparing the hydrodynamic model, empirical approach, and observed vertical profiles for January, February, and March of 2015 are presented in Figure 10. The hydrodynamic model values and the observed temperature were obtained by Coelho, Almeida, and Mateus [29] for the water quality assessment of Aguieira Dam. For this study, they used the model CE-QUAL-WE [30], which is a laterally-averaged water-quality and hydrodynamic model recommended by the United States Environmental Protection Agency (EPA) for comprehensive two-dimensional water quality studies. In CE-QUAL-W2, the laterally-averaged three-dimensional continuity and momentum (conservation of the fluid mass and conservation of momentum, respectively) equations that govern reservoir hydrodynamics are resolved numerically using finite difference methods.

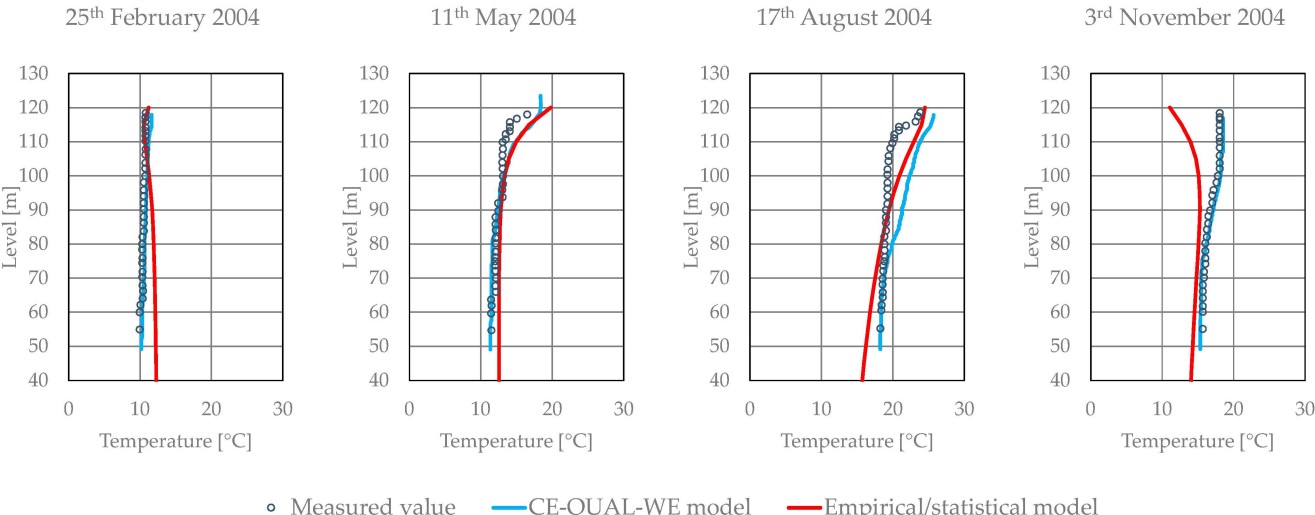

**Figure 10.** CE-QUAL-WE [29] and empirical/statistical models versus the measured temperature [29] profiles upstream of Aguieira Dam in 2004.

Except on 3 November 2004, the empirical model gives reasonably good results. The deviation observed in November is due to the imposed harmonic behavior adopted in Equation (30). In fact, as pointed out by Sun et al. [31], for the numerous functions of flood control, power generating, water supply, and navigation, dams are operated according to a seasonal variation plan, such as retaining water at the end of the wet season and releasing it in the dry season. As a result of such operation, the annual thermal cycles

may exhibit seasonally varying phase delays, and the seasonally varying temperature fluctuation may no longer follow a harmonic variation. Nevertheless, due to the advantage of rapid calculation time and no need for complex data, the empirical model is preferred for thermo-mechanical analysis in contrast to the CE-QUAL-WE model.

## 9. Analysis and Results

The transient thermal analysis was performed with a new version of the in-house code PAT [3]. The analysis was conducted considering an incremental time of 1 h.

Regarding the boundary conditions, convection and atmospheric and solar radiation actions were considered at the exposed surfaces of the dam; fixed reservoir water temperature was applied in all submerged boundaries. For the rock mass foundation, convection and atmospheric radiation boundary conditions were applied in all air-exposed boundaries; fixed reservoir water temperature boundary conditions were applied in all submerged boundaries; adiabatic boundary conditions were considered at the lateral boundaries; a fixed temperature boundary condition of 13 °C was imposed at the bottom. The last temperature was fixed as the mean air temperature minus 2 °C.

It is worth noting that the limit between convection and fixed reservoir water temperature boundary conditions must be in accordance with the discretization adopted for the problem. This is because the integrals containing the convection coefficient $h_t$, i.e., Equations (13) and (14), are integrated over the whole face of the element where the convection boundary condition is applied.

It is also important to note that due to the presence of a convective component in the heat stiffness matrix, Equation (13), the application of variable reservoir water level implies the necessity of updating this matrix each time the water level covers different elements. Since this procedure is time-consuming, a constant reservoir water level of 120 m was adopted.

In order to validate the model, a comparison between the predicted and measured temperatures was made for five groups of thermometers. Each group is formed by three thermometers located at the downstream face, in the middle of the thickness, and at the upstream face, respectively. The first three groups are installed in the middle of the central arch at elevation 53 m (thermometers T3, T5, and T7), 75 m (thermometers T27, T29, and T31), and 120.20 m (thermometers T58, T60 and T62). The fourth group is located in the right arch at elevation 65 m (thermometers T15, T17, and T77), and the fifth group is located in the left arch at elevation 65.20 m (thermometers T10, T12, and T14).

Figures 11–15 show the comparison between recorded temperatures and those predicted with the numerical model, both with and without considering the shaded zones of the dam. In all graphs, the results neglecting the shadow effect (yellow line) were represented at the back of the curve corresponding to the calculation with the shadow effect (grey line). In these calculations, the solar radiation was estimated using the exponential function given in expression (27).

The analysis of these figures reveals that both models, the one without shadow effect and the one with shadow effect, achieve a good agreement with the measured temperatures. Only in the cases where the solar radiation was occluded by the dam's own geometry at the main hours of sunshine was the overall computed temperature at the downstream surface higher for the model without shadow, thermometers T3, T10, and T15.

In general, there is a good agreement between the estimated and the measured temperatures at the upstream face of the dam. Nevertheless, this conclusion must be taken cautiously because the model was constructed based on these measured temperatures.

For the thermometers located in the middle of the sections, a very good agreement between the measured and estimated values was achieved.

Finally, Figure 16 compares the results obtained with the two radiation models presented in Section 7.5. As can be observed, both models provide similar results.

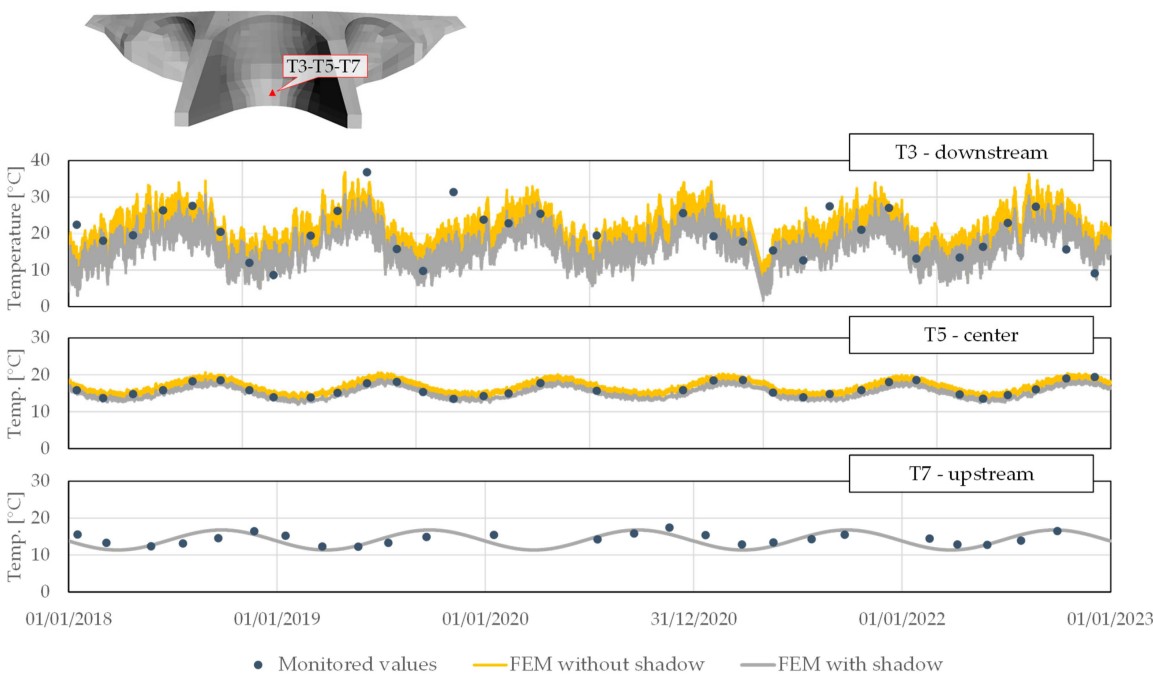

**Figure 11.** Comparison of the predicted and monitored temperatures at thermometers T3, T5, and T7.

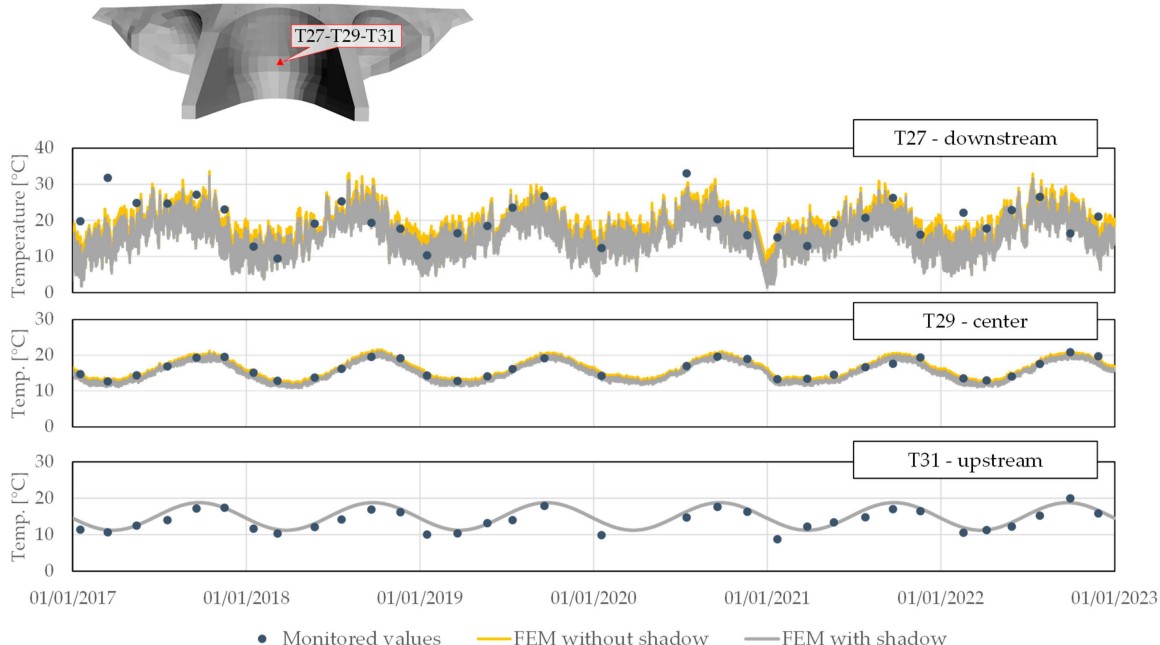

**Figure 12.** Comparison of the predicted and monitored temperatures at thermometers T27, T29, and T31.

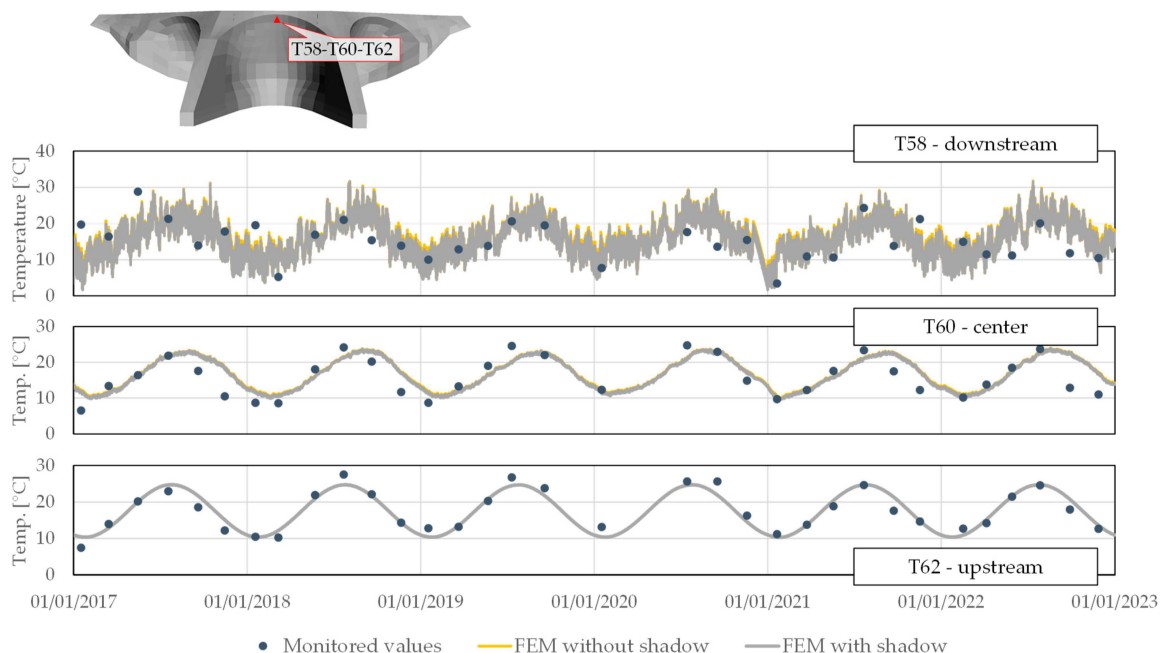

**Figure 13.** Comparison of the predicted and monitored temperatures at thermometers T58, T60, and T62.

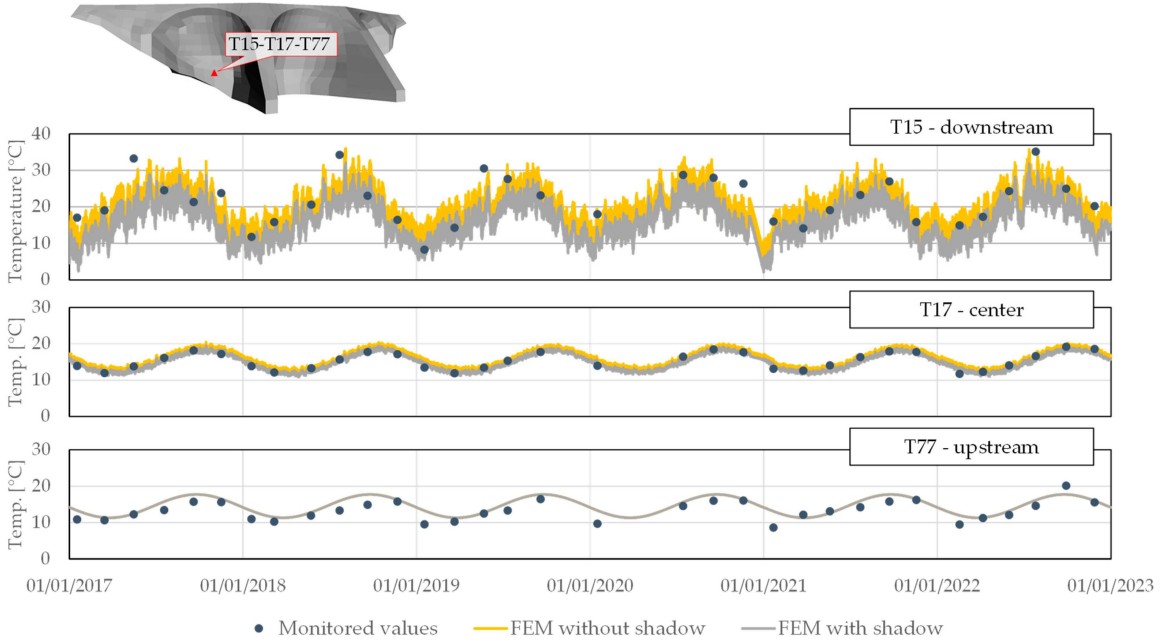

**Figure 14.** Comparison of the predicted and monitored temperatures at thermometers T15, T17, and T77.

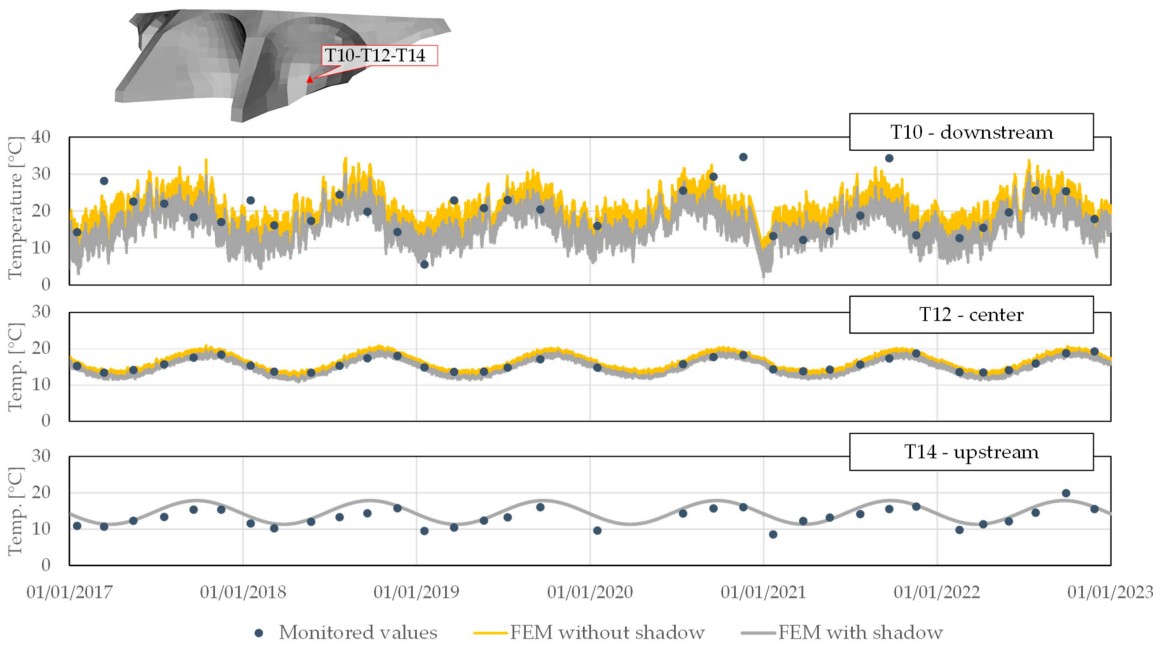

**Figure 15.** Comparison of the predicted and monitored temperatures at thermometers T10, T12, and T14.

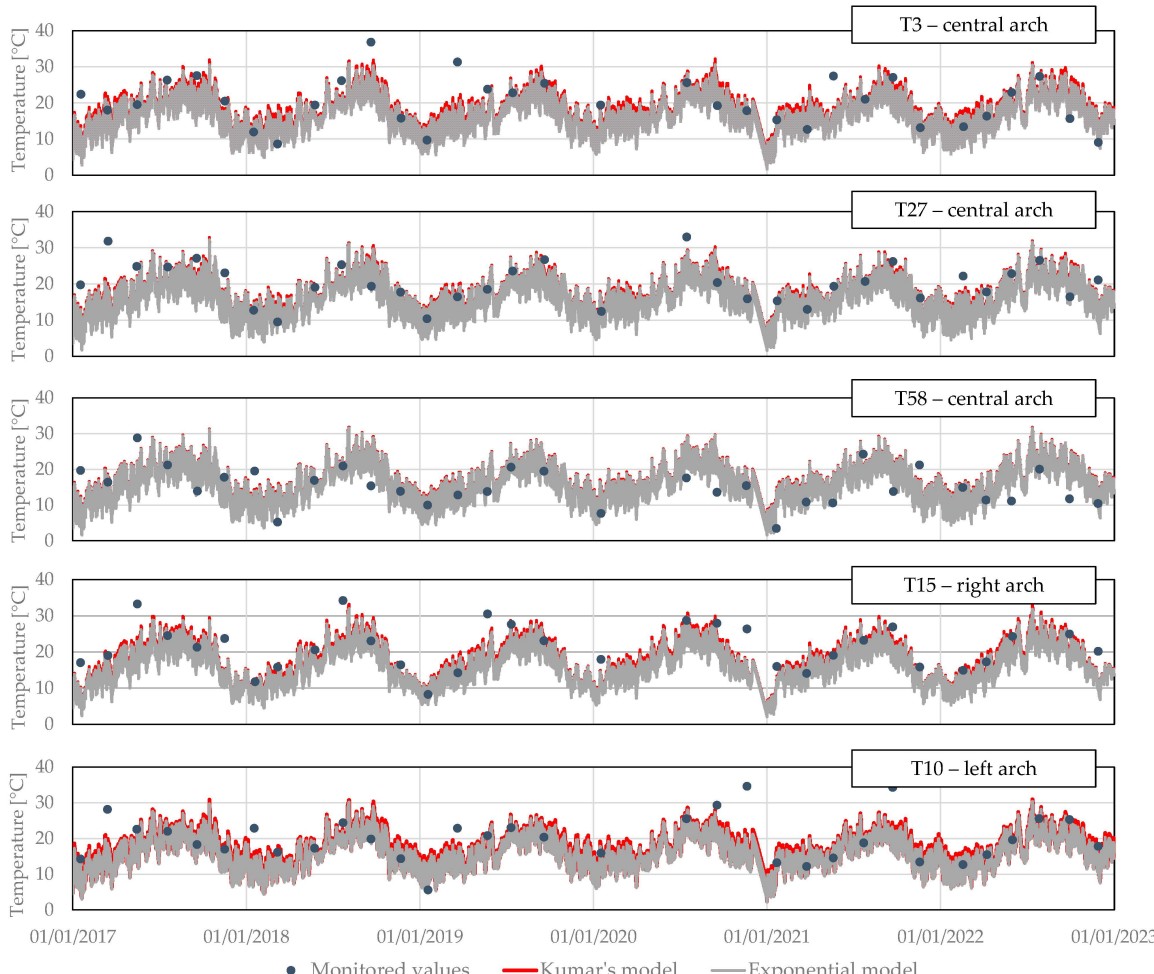

**Figure 16.** Comparison of Kumar's model and the exponential function.

## 10. Discussion

As mentioned before, the objective of the coupled thermo-mechanical analysis, the age of concrete, and the required level of agreement between reality and simulation outcomes will determine the degree of complexity of the thermal model. Moreover, and not less important, is the time spent in the simulation. For this reason, in this study, some decisions, such as the level of refinement of the mesh and the fixed convective boundary conditions, had been taken beforehand.

In relation to the level of refinement of the mesh, it always results from a compromise between accuracy and computational cost. In fact, the mesh of the Aguieira Dam corresponds to the model actually used to assess the monitored behavior of the dam; therefore, the level of refinement was adopted in order to return the thermo-mechanical analysis results in an acceptable period.

Regarding the convective boundary conditions, the changes related to the reservoir water level fluctuation or the simple variation of the convection heat transfer coefficient caused by wind velocity have been neglected using fixed values. This simplification is due to code PAT using a direct solver for the system of equations. Therefore, any change in the matrix would require a new factorization, which is the most time-consuming step in solving the system of equations.

The constraints above affected mainly the downstream surfaces of the dam. On the one hand, the lower refinement in thickness leads to a poorer representation of the diurnal temperature variation. The "rigidity" imposed by the low level of discretization brought down the difference between daytime and nighttime temperatures.

On the other hand, the fixed convection heat transfer coefficient did not reflect the actual condition of the wind velocity.

Regarding the fixed water level, this approximation did not strongly influence the results. In all cases, the major source of disagreement between the monitored and estimated values was related to the empirical/statistical model adopted for the water temperature.

Without losing sight of the above limitations, the results obtained suggest that the shadow effect is more important at lower levels because, at higher levels, the slope of the surface blocks the sun's rays at the main hours of sunshine, as shown in Figure 6. Nevertheless, even at lower levels, such as the location of thermometer T3, the maximum difference in temperature between considering or not the shaded area is 6.9 °C, with an average value of 3.0 °C. As expected, this difference in temperature attenuates away from the surface, and at the location of thermometer T5, the maximum difference between considering or not the shaded area is 3.0 °C, with an average value of 1.5 °C.

Although the two solar radiation models present major differences in their developments, it can be seen from Figure 16 that they do not reveal a big discrepancy in results. In this sense, it is important to note that the exponential model, calibrated using the data measured in a nearby solar radiation station, considers only the beam radiation component, while Kumar's model computes both the beam and the diffuse radiation components. Moreover, Kumar's model is based on semi-empirical expressions formulated by Germeles in 1966 using the data of the solar energy absorbed by the earth's atmosphere measured in different locations in the USA [32].

Regarding the reservoir water temperature, the greater lag between estimated and monitored values occurs at lower levels. This is due to the effect of the reversible pump-turbine units (see Figure 2), which, either in pumping mode or in generating mode, influence the water temperature due to the water inflow or outflow, respectively. Figure 9 already anticipated that the parameter representing the phase difference of water temperature was the parameter that presented the greatest dispersion.

Overall, it is clear that the increase in the sophistication and complexity of the thermal analysis does not translate in a linear fashion into improvements in the results. In fact, the level of detail in representing the spatiotemporal fields affects the behavior near the boundary where they are applied. Therefore, the level of complexity will depend on the

purpose of the analysis. The overall performance of the dam will always be less demanding than the evolution analysis of the concrete cracking.

In the end, it can be concluded that the selection of the main phenomena involved in the thermal analysis is a matter of engineering judgment and experience. Nevertheless, the creation of a workgroup of experts that produce guidelines, technical reports, or implementation guides about the phenomena involved in the thermal analysis of large concrete dams would be very useful to the dam community.

**Author Contributions:** Conceptualization, N.S.L. and S.O.; methodology, N.S.L.; software, N.S.L.; validation, N.S.L. and S.O.; formal analysis, N.S.L. and S.O.; investigation, N.S.L.; resources, N.S.L. and S.O.; data curation, N.S.L. and S.O.; writing—original draft preparation, N.S.L.; writing—review and editing, N.S.L. and S.O.; visualization, N.S.L.; supervision, N.S.L. All authors have read and agreed to the published version of the manuscript.

**Funding:** This research received no external funding.

**Data Availability Statement:** The software used in this study is available at https://github.com/nschclar/PAT (see folder PAT_V1_3, uploaded on 27 August 2023, for the latest version of PAT). The finite element mesh is not publicly available due to confidential constraints.

**Acknowledgments:** The authors thank the owner of the dam, EDP—Gestão da Produção de Energia, S.A.

**Conflicts of Interest:** The authors declare no conflict of interest.

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
