# Peer review of "Insights about Modelling Environmental Spatiotemporal Actions in Thermal Analysis of Concrete Dams: A Case Study"

_2673-7264, doi:10.3390/thermo3040036_

Round 1

Reviewer 1 Report

1.      The value of the absorption coefficient "a", Equation (9), is not defined in the paper

2.      In Equation (14), qr needs to be changed with qq

3.      In lines 146 and 147, it is written "the prescribed heat flow and the third integral takes into account the convection heat transfer", and it should be "the solar radiation heat flow and the third integral takes into account the convection and radiation heat transfer"

4.      In line 180, it is written “kg/m3”, and it should be "kg/m3

5.      In line 184, it is written “W/(m2 K)”, and it should be "W/(m2 K)”

6.      In line 575, it is written “2000 to 201”, and it should be "2000 to 2015”

Reviewer 2 Report

Please, see attached file.

Reviewer 3 Report

The article "Insights about modelling environmental spatiotemporal actions in thermal analysis of concrete dams: A case study" presents the thermal analysis of a concrete multiple arch dam in order to evaluate and validate the spatiotemporal representations used to model solar radiation and water temperature boundary conditions.

The paper is clearly written and structured and is worth publishing after some minor improvements:

1. The novelty/originality of the study is not clearly stated. There are several papers dealing with spatiotemporal environmental actions in the thermal analysis of concrete dams. What does this paper contribute to the state of the art? Please clarify in the introduction.

2. In line 393, why has 13°C been chosen?

3. The way of calculating solar radiation is very interesting and well explained, however, as you point out in the manuscript, it is also very time-consuming. It would be nice to include in the comparison an other simplified calculation, for example, increasing the daily mean temperature to consider the effect of solar radiation (as in [1]).

If carrying out new calculations is too costly, at least explain the advantages of considering the solar radiation in the calculation. In which cases is it more advisable to take it into account?

Round 2

Reviewer 2 Report

In improved version some language deficiencies may be found (e.g. L115 "directions" - plural, and others). Authors should revise the language in the text.
